# Fatal Clostridium Infection in a Leg-Amputated Patient after Unsuccessful Knee Arthroplasty

**DOI:** 10.3390/ijerph18179186

**Published:** 2021-08-31

**Authors:** Enrico Maria Zardi, Paolo Persichetti, Alessio Palumbo, Edoardo Franceschetti, Francesco Franceschi

**Affiliations:** 1Internistic Ultrasound Service, Campus Bio-Medico University, 00128 Rome, Italy; 2Department of Plastic, Reconstructive and Aesthetic Surgery, Campus Bio-Medico University, 00128 Rome, Italy; p.persichetti@unicampus.it; 3Department of Orthopaedic and Trauma Surgery, Campus Bio-Medico University, 00128 Rome, Italy; a.palumbo@unicampus.it (A.P.); e.franceschetti@unicampus.it (E.F.); 4Department of Orthopaedic and Trauma Surgery, San Pietro Fatebenefratelli Hospital, 00189 Rome, Italy; f.franceschi@unicampus.it

**Keywords:** amputation, antibiotic, clostridium infection, prosthesis, skin wound, therapy

## Abstract

Prosthetic joint infection (PJI) is a possible complication occurring after prosthesis implantation. We describe the case of a patient with early postoperative multidrug-resistant polymicrobial PJI and mixed infection of the surgical wound. Despite the removal of the prosthesis, the positioning of double-stage exchange, and dehiscence debridement of the surgical wound, the infection continued. Positioning of an external fixator, plastic reconstruction with a skin graft, and continuous (two years) multiple antimicrobial therapy led to the resolution of the knee infection; a knee prosthesis was implanted, but a new infection of the extensus apparatus by multidrug-resistant *Klebsiella pnumoniae* followed. It was complicated by surgical wound dehiscence, forcing us to remove the prosthesis, put a new external fixator, and continue with the antibiotic treatment, with no results, and, finally, proceed to a leg amputation. Fourteen days after, the patient was discharged in good clinical condition but, fifteen days later, during rehabilitation in another hospital, the patient developed a severe *Clostridium difficilis* infection with profuse, intense diarrhea, toxic megacolon, and septic shock; despite colectomy and treatment in an intensive care unit, he died four months later. Patients affected by polymicrobial PJI are at high risk of treatment failure and, therefore, should be given a warning, in good time and appropriate form, of the likelihood of leg amputation.

## 1. Introduction

Prosthetic joint infection (PJI) is a complication occurring in 1.4–2.5% of arthroplasties [1]. In most cases, we are dealing with monomicrobial PJIs since polymicrobial PJIs account for 4–19% of cases [2,3], rising to 37% if developed in the early postoperative period [4].

PJI starts when a pathogen, accidentally introduced into the joint, binds to the surface of the prosthesis by means of adhesion molecules; then, thanks to the production of biofilm, the pathogen protects itself from the antibiotic action and strengthens the bond with the implant [5]. The formation of polymicrobial biofilm is an even more serious event that makes any type of treatment more difficult [5].

Debridement, antibiotics, irrigation, and implant retention (DAIR) may be considered effective treatment options for early PJI [6] under some specific conditions: a stable prosthesis, easily treatable pathogen, and absence of a sinus tract [7].

Revision surgery is another important option to treat PJI, but this is burdened by an infection rate over 10% [8], which is not without clinical impact. Indeed, it may raise the risk of prolonged hospitalization and duration of antibiotic therapy at home, increasing costs and causing relevant social and economic problems.

Here, we illustrate a multidrug-resistant polymicrobial PJI of the knee, complicated by a mixed infection of the surgical wound, which broke out within a week of the prosthetic intervention.

## 2. Case Report

A 76-year-old overweight man (body mass index of 29) with osteoarthrosis, who had already successfully undergone left hip and knee surgeries some years previously, also underwent right knee arthroplasty in another hospital, developing polymicrobial PJI (*Enterococcus spp*, *Klebsiella pneumonia*, *Staphylococcus aureus*), within a few days of surgery. Since the conditions to perform DAIR were not met, he was treated with two-stage exchange and antibiotic therapy (daptomycin and meropenem), suffering side effects (such as anemia and leukopenia) and developing carbapenem resistance. After one year of continued antibiotic treatment with glycopeptides, cephalosporin, and quinolones, a knee prosthesis reimplantation was attempted without success, being complicated by patellar tendon rupture, mixed infection of the wound, and dehiscence. As anemia, leukopenia, and high serum inflammatory markers still persisted and wound dehiscence did not heal with negative pressure wound therapy, the patient was admitted to our hospital. On admission, he was tired, pale, had knee pain that was treated with *non-steroid* anti-inflammatory drugs, and his inflammatory markers were increased. Laboratory analysis showed: erythrocyte sedimentation rate (ESR) 79 mm/h (normal value <37mm/h), C reactive protein (CRP) 13.6 mg/dL (normal value <0.5mg/dL), white blood cells count 3550 cells/µL, neutrophils count 1750 cells/µL (49%), red blood cells count 3,160,000 cells/µL, hemoglobin 11.1 g/dL, platelets count 286,000/µL, creatinine 0.83 mg/dL. His American Society of Anesthesiologists (ASA) score was 2. The prosthesis was promptly removed, and an antibiotic-impregnated spacer was put. Antibiotic therapy was adjusted according to the microbial cultures collected from (a) the knee surgical wound (Figure 1A) (positive for *Staphylococcus hominis*) and (b) the knee joint (positive for extended-spectrum β-lactamases (ESBL) *Klebsiella pneumoniae*, and multidrug-resistant (MDR) *Acinetobacter baumannii*, only sensitive to colistin). Tigecycline (50 mg every 12 h), daptomycin (500 mg a day), and colistin (4.5 MU every 12 h) were started. Due to the progressive worsening of renal function (creatinine values oscillating between 1.32 and 2.8 mg/dL), the dosage of the latter two antibiotics was gradually reduced within one month to 300 mg a day and to 2 MU every 12 h, respectively, based on estimated creatinine clearance. Rifampin was also orally administered two hours after meals at a dosage of 600 mg a day. Iron deficiency and anemia were corrected with intravenous iron load, intramuscular vitamin B12, erythropoietin administration (10.000 IU twice a week), and, when necessary, red blood cell transfusions (if hemoglobin was less than 7 g/dL). Vitamin D deficiency and low potassium and magnesium were also restored. Reconstructive surgery of the knee wound was attempted, and 40 cycles of hyperbaric treatment were performed with only partial benefit. A number of microbial cultures collected from (a) the knee wound and (b) the knee joint during the change of the first spacer with another one were positive for *MDR Pseudomonas aeruginosa* (sensitive only to colistin) and *Staphylococcus capitis*. Inflammatory markers were increased: ESR 52 mm/h, CRP 1.10 mg/dL, D-dimer 1762 ng/mL (normal value <500 ng/mL). Other laboratory analyses showed: white blood cells count 2780 cells/µL, neutrophils count 1290 cells/µL (46%), red blood cell count 2,100,000 cells/µL, hemoglobin 7.2 g/dL, platelets count 216,000/µL, creatinine 1.83 mg/dL. Tygecicline and daptomycin were discontinued, while an antibiotic combination (ceftolozane/tazobactam, 500 mg/250 mg in 100 mL of saline (0.9%) solution intravenously three times a day, according to estimated creatinine clearance) was given in addition to colistin and rifampin; the latter were stopped after two months because of the persistence of high serum inflammatory markers (ESR 60 mm/h, CRP 1.21 mg/dL, D-dimer 1598 ng/mL), while fosfomicin (4 g intravenously, three times a day) was added to ceftolozane/tazobactam, with benefit against *Staphylococcus capitis*. Due to the persistence of MDR and carbapenem-resistant *Pseudomonas aeruginosa* in cultures collected from the knee wound, the spacer was removed, an external fixator was put in, and a daily antibiotic treatment was continued in a rehabilitation center. Three months later, healing of the knee wound (Figure 1B) and normalization of all serum inflammatory markers occurred. During this period, *Corynebacterium striatum* was isolated from a skin purulent fluid collected near the fixator pin of the thigh and treated, with benefit, by administering dalbavancin at a dosage of 1.5 g intravenously (two times, one week apart). After a further three months, thanks to the persistent normal values of serum inflammatory markers, ceftolozane/tazobactam and fosfomicin were stopped; two months later, the patient underwent labeled leucocyte scintigraphy and positron emission tomography, which excluded bone and joint pathogenic processes. Hence, the patient underwent new right knee arthroplasty. Some days after implantation, there was dehiscence of the surgical wound, and MDR *Klebsiella pneumoniae* was isolated from the surgical wound. The patient had fever and showed a decrease in blood pressure. Blood analysis showed electrolyte disturbances (potassium 1.8 mmol/L (normal value 3.5–5.1 mmol/L), chlorine 85 mmol/L (normal value 98–107 mmol/L), calcium 7.5 mg/dL (normal value 8.4–10.2 mg/dL)), phosphorus 1.7 mg/dL (normal value 2.3–4.7 mg/dL), and high serum inflammatory markers (ESR 51 mm/h, CRP 5.29 mg/dL, D-dimer 1070 ng/mL). A solution for intravenous infusion of 500 mL glucose (5%) combined with potassium 80 mEqs was promptly administered, and calcium and phosphorus were restored orally. Although blood culture and microbial culture collection from the knee joint were negative, antibiotic treatment was started using an antibiotic combination (ceftazidime/avibactam 2/0.5 g in 100 mL of saline (0.9%) solution, administered intravenously three times a day; after a few days, reduced to 1/0.25 g, three times a day) and fosfomicin (4 g in 100 mL water for injection, administered intravenously three times a day). After one week, the clinical conditions of the patient improved; there was normalization of electrolyte balance and reduction of serum inflammatory markers (ESR 34 mm/h, CRP 1.09 mg/dL, D-dimer 830 ng/mL). Unfortunately, the wound showed signs of necrosis; a tomography revealed severe cellulitis of soft tissue near the prosthesis and, during surgical curettage, a severe detachment of the skin, extending over several cm^2^; the presence of a fistula was noted. The prosthesis was removed, a new external fixator was put in, and, after fifteen days, due to the worsening of the wound, leg amputation was performed (see Figure 2). After two weeks, the wound of the amputation had healed, and he was transferred in good clinical condition to a clinical rehabilitation center. Fifteen days later, the patient developed a severe infection from *Clostridium difficile*, which quickly caused profuse, intense diarrhea and, after two days, toxic megacolon and septic shock despite treatment with oral vancomycin (125 mg orally, four times daily) and metronidazole (500 mg, three times daily). Therefore, he was urgently transferred to an intensive care unit of a general hospital, where vancomycin and metronidazole were given via rectal Foley and intravenously (500 mg in 100 mL normal saline solution every 6 h and 500 mg every 8 h, respectively) and supplemental oxygen, intravenous fluids (albumin, balanced salt solutions), medications (particularly vasopressors), and blood products were also promptly administered without success, thus making it necessary to perform a colectomy. The postoperative course was complicated by electrolytic disturbances, anemia, acidosis, and malnutrition, which were treated with salt solutions, red blood cells transfusions, bicarbonate, and nutrition support; however, after four months of intensive care, he died.

## 3. Discussion

We described an early postoperative multidrug-resistant polymicrobial PJI of the knee, complicated by a surgical wound infection. Despite all attempts to find a long-term surgical and medical cure, our patient’s leg had to be amputated; after discharge to a clinical rehabilitation center, he became infected with a fatal *clostridium* infection. There was no benefit from long-term antibiotic and surgical treatments that are considered viable options to treat monomicrobial PJI [9] but may fail to treat polymicrobial infections.

Indeed, polymicrobial PJI is an independent risk factor for infection recurrence [10] and has a reduced cure rate when compared with that of monomicrobial PJI [11]. Despite the efficacious use of spacer exchange, in these cases, other therapeutic strategies should become part of the common lexicon, such as leg amputation [5].

Polymicrobial infection, observed in 21–37% of patients with early postoperative PJI [4,12], as is the case of our patient, may be complicated by antibiotic resistance and is considered one of the major problems to be managed now and even more in the future [13].

To prevent this scenario, a change in treatment management should be planned; old and new antibiotics should be used with thriftiness and in a targeted manner based on microbiologic investigations, avoiding the selection of mechanisms of resistance [14,15].

The change of the antimicrobial susceptibility pattern in 60% of persistent microorganisms between the first and second stages of the procedure severely complicates the management of the infection [16], contributing to an increase in antibiotic resistance. When the patient came to our attention, having been affected by polymicrobial PJI and after two unsuccessful attempts of prosthesis implantation, we started long-term targeted antibiotic therapy along with surgical treatments, and the patient healed.

The combination of antibiotic therapy (ceftolozane/tazobactam and fosfomicin) and the use of an external fixator turned out to be a winning strategy to obtain the eradication of the infection.

However, after positioning a new knee prosthesis, despite all measures to avoid infections, the patient was infected with PJI again, with no improvement despite clinical and surgical cures, the removal of the prosthesis, and the positioning of another external fixator. In these cases, leg amputation is needed, and this was performed without any complication; the patient was discharged in good clinical condition. How the patient was infected with clostridium infection during hospitalization in a rehabilitation center (with toxic megacolon and septic shock, although he never had diarrheal episodes in the past) is disputed.

*C. difficile* is a major cause of healthcare-related infection that begins through the production of toxins A and B [17]. Antibiotic selective pressure and resistance can disrupt normal host intestinal microbiota; in addition, *C. difficile* can be found in meat products, seafood, and salads, and the widespread environmental dispersal of its spores can also play a role [17]. Which one of these causes has favored the insurgence of such violent *C. difficile* infection is not clear; we cannot rule out a combination of these causes.

## 4. Conclusions

We highlight that PJI may be favored by the presence of patient-related risk factors, such as the presence of sociodemographic characteristics, elevated BMI, and history of previous surgery, as occurred in this case. The low resistance of human tissue to tolerate, in elderly age, repeated surgeries in a relatively short period of time may have contributed to the unsuccessful outcome.

Moreover, we reaffirm the difficulties in managing polymicrobial PJIs, especially in patients who have already undergone previous prosthesis implantations. They deserve to be informed promptly about the likelihood of leg amputation since the success rate of multiple revision surgeries, endoprosthetic joint reconstruction, and limb salvage ranges between 43% and 62% only [18], with about half of the cases doomed to failure.

Therefore, it is important to implement the most accurate preventive strategies by adjusting the antibiotic prophylaxis regimen and improving disinfection and postoperative wound care whenever any patient undergoes surgical procedures [19].

These behaviors, together with all measures to prevent *C. difficile* infection, including those suitable for identifying healthy carriers [20], should be adopted, especially in the case of patients undergoing long-term antibiotic treatments.

## Figures and Tables

**Figure 1 ijerph-18-09186-f001:**
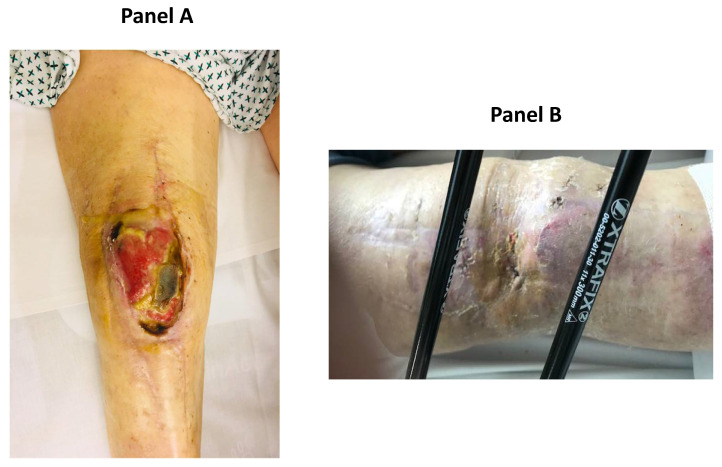
**A**: Photograph demonstrating septic surgical wound of the right knee after removal of the prosthesis (by S. Hominis). **B**: Photograph demonstrating the wound healing of the right knee after three months from the positioning of the external fixator.

**Figure 2 ijerph-18-09186-f002:**
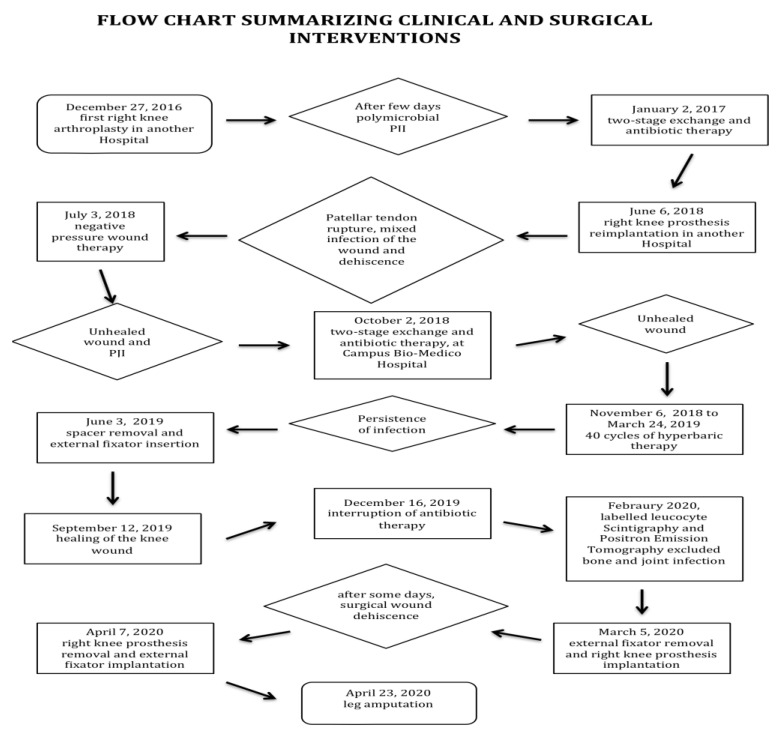
Flow chart summarizing clinical and surgicalinterventions.

## Data Availability

No new data were created or analyzed in this study. Data sharing is not applicable to this article.

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
