# Peer review of "Fatal Clostridium Infection in a Leg-Amputated Patient after Unsuccessful Knee Arthroplasty"

_ijerph, 2021, doi:10.3390/ijerph18179186_

Round 1
Reviewer 1 Report
Dears,
Thank you for the opportunity to review the manuscript entitled "Fatal clostridium infection in a leg amputated patient after unsuccessful knee arthroplasty". Please find my reviewer comments below:
- I suggest a review of the text by a professional English proofreading service or a native English speaker.
- The sentence on lines 38-39 is unclear. Please consider to re-write.
- I propose to remove the sentence on lines 49-51.
- The sentence on lines 56-62 is quite long. Please consider to re-phrase.
- Abbreviations that are only used once are not necessary, e.g. “VAC” and “NSAID”. Please consider to remove.
- Although common, “VAC” is an abbreviation used by one company. Please consider changing to “negative pressure wound therapy (NPWT)”.
- Please be consistent with how you use capitalization, e.g. “red blood cells count” and “Platelets count” on line 92.
- The sentence on lines 155-158 is quite long. Please consider to re-phrase.
Reviewer 2 Report
It is a very interesting case study. Two suggestions to help the reader understand the implications of this report:
(1) Whether the authors can relate the patient to the incidence that the authors had summarized at the beginning, e.g. whether the conditions of the patient are fitting the 4-19% of the cases reported by ref. 2,3. Or how the patient individual conditions are differing from the reported studies.
(2) English writing should be checked more thoroughly. Some places the text is either colloquial or misleading/incorrectly written. Some conventions should be adapted e.g. Line 80 "oral rifampin was also administered" --Rifampin was also orally administered. line 81 "away from meals" is the oral administration after meals or with or along with the meals?
Reviewer 3 Report
This case report is about course of events after total knee replacement in 76 years old overweight patient who suffered a few days after surgery polymicrobial prosthetic joint infection (PJI). After additional surgeries two revisions, two external fixation attempts and finally amputation, patient suffered from Clostridium infection and died. This case report is presenting rather complicated course of surgical and medical interventions. It should be suggested that a flow-chart (time-table) of course interventions with date and name of interventions is included in the manuscript so it could be easier to follow. This is for benefit of readers. Also, one fact is missing from the manuscript. It means, that exact period from the time when PET scan and labelled leukocyte scintigraphy has proved that there is no infection in the knee joint to the time of subsequent total knee revision is not stated. If this period is too short, this could explain subsequent infection after three attempts of total knee replacement implantation.
In discussion section, it is suggested that some comments should be given to possible underestimation of low resistance of human tissue capacity in elderly age for repeated surgeries in relatively short period of time, in previously infected environment. Another point of interest is missing, i.e. vascularity status of lower limb where repeated surgical attempts were performed. It would be of interest to know what is standpoint of authors as per double check to prove that polymicrobial infection is eradicated.
